# VisHanfu: An Interactive System for the Promotion of Hanfu Knowledge via Cross-Shaped Flat Structure

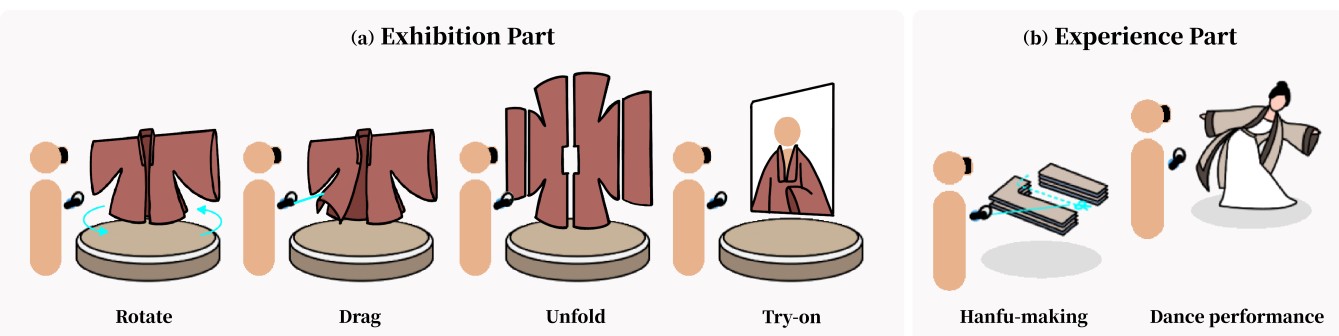

**Figure 1: VisHanfu is an interactive virtual reality system designed to promote Hanfu culture. Within the "Exhibition part" (a), users have the opportunity to observe the digital restoration model of Hanfu artifacts with four interactive options: Rotate, Drag, Unfold, and Try-on. In the "Experience part" (b), users can actively participate in the Hanfu-making process and view the fluid movements of the Hanfu they create, which involves simple actions like dragging and dropping, alongside watching a dance performed by an avatar.**

## ABSTRACT

Hanfu is the representative traditional costume of Han nationality in China, which carries the outstanding craftsmanship of dyeing, weaving, and embroidery, and is of great significance to the inheritance of traditional culture. However, the existing methods of Hanfu publicity still have problems, which are not conducive to the inheritance of Hanfu culture. In this work, we developed the VisHanfu virtual reality system by focusing on the "Cross-Shaped Flat Structure", which is an integral feature of Hanfu. We have digitally restored twenty-five representative Hanfu historical artifacts and provided an interactive making experience. Combined with high realistic cloth simulation techniques, it allows users to interactively observe the movement effects of the Hanfu. The results of user studies demonstrates that our system can provide a favorable experience for users, and bring a better learning effect, which helps users to enhance their interest in learning and thus contributes to the inheritance of Hanfu culture.

## CCS CONCEPTS

• **Human-centered computing** → **Virtual reality**; **User studies**.

## KEYWORDS

Hanfu, Cultural Heritage, Virtual Reality, Interactive Design

**Unpublished working draft. Not for distribution.**

## 1 INTRODUCTION

Hanfu, which encompasses all the traditional clothing categories of the Han nationality in China, has a recorded history of more than 3,000 years [8]. It is a demonstration of the distinguished craftsmanship and aesthetics of Chinese dyeing, weaving, and embroidery, associated with a number of Chinese intangible cultural heritages such as Su Embroidery [23] and silk craftsmanship [43], which are of great significance in preserving the traditional culture of China. Cross-shaped flat structure (Figure 2) is the typical structure of Hanfu, which greatly distinguishes it from other ethnic groups and modern costumes. Its main features can be summarized in three points [41]: (1) "Cross-shaped": when laid flat, the lines of the shoulders and sleeves form a cross-axis with the front and back center lines; (2) Unity: The body and sleeves are sectioned from the fabric while keeping the front and back parts connected, and the whole piece of fabric is directly utilized for splicing; and (3) Flatness: traditional Hanfu is not based on three-dimensional tailoring design. These characteristics are closely related to the frugal spirit of the ancients, who wished to maximize the use of fabrics and avoid waste.

With the emphasis on traditional culture in recent years, Hanfu, as an important part of it, has also received increasing attention. The results of a survey [34] in which 1,902 participants were asked to report how they learned about Hanfu culture show that 59.3% of them through film and television productions, 54.7% through social media, and the next most popular sources were Hanfu cultural club (49.1%), historical documents (40.7%), and museum exhibits (38.8%). However, there are still some problems with the existing methods of publicizing Hanfu culture for the general public: (1) The way of associations and museum exhibitions requires people to be physically present, which is not convenient enough and is not friendly to people in such areas where cultural resources are not rich enough; (2)

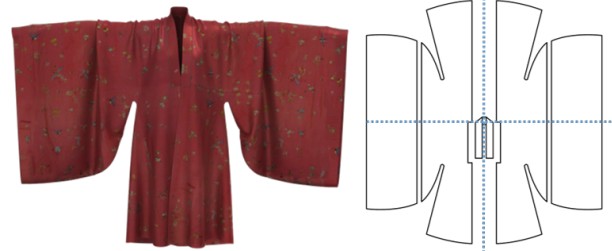

**Figure 2: The Cloak with Floral and Bird Embroidery[27] and its structure diagram. The cross-shaped flat structure is illustrated in blue dotted line**

Although the way of books and videos is relatively convenient, the audience can often only passively accept the author's output, and it is difficult to bring personalized and immersive learning experience for users; (3) Some popular movies and TV shows contain fallacies, for example, the costumes worn by the actors do not match their historical backgrounds, bringing misinformation to the audience. Additionally, Hanfu is also a commodity with a sizable market; in China alone, it has a market worth over billions of dollars and millions of consumers [18]. However, Hanfu sold on the market are frequently modified styles in order to satisfy the aesthetic and daily wear needs of people. This makes it difficult for the public to understand the traditional Hanfu knowledge of structure, texture, and its corresponding cultural connotations and may even cause confusion. For instance, some merchants sell the tops and bottoms from different dynasties in sets.

To address the above problems and to properly publicize Hanfu culture, we implemented a virtual reality system, VisHanfu, focusing on the cross-shaped flat structure of Hanfu. In the system, we introduced the formation reasons and characteristics of the cross-shaped flat structure, and selected twenty-five representative Hanfu artifacts of different dynasties to be digitally restored for users to interact. Additionally, the VisHanfu system offers an interactive and easy-to-use making functionality that enables users to more naturally experience the process of cutting, stitching, and sewing Hanfu. Combined with the high realistic cloth simulation technology, users can observe the non-rigid motion effect of the Hanfu they created by dragging and dropping the clothing model by operating the controller. We also invited an experienced dancer to perform a Chinese dance, recorded her movement data, and migrated it to the MetaHuman [14] model so that users could watch an avatar dance while wearing the Hanfu they made in the system. We deployed the VisHanfu system in a local museum for a week, invited 150 visitors to experience it, and conducted a laboratory user study containing 60 participants. The results indicate that our system helps users enhance their interest in learning about Hanfu culture and answer questions more correctly after learning with our system than learning by reading paper materials.

The main contributions of our work include:

(1) We developed VisHanfu, an interactive virtual reality system focusing on the cross-shaped flat structure, to overcome the lack of convenience, immersion, and fun in existing Hanfu culture dissemination methods, and to help the promotion of Hanfu culture;

(2) A digital restoration scheme for Hanfu artifacts was proposed, which improves the efficiency of texture reconstruction for Hanfu, and twenty-five famous Hanfu artifacts was reconstructed;

(3) Combined with high realistic cloth simulation technology, the non-rigid movement of Hanfu model in virtual reality is realized, allowing users to observe the movement effect of the clothes in real time through simple interactions such as dragging and dropping, which provides insights for enriching the user's interaction experience with Hanfu and other clothing artifacts.

## 2 RELATED WORK

### 2.1 Virtual Reality Exhibition

As Virtual Reality (VR) technology advances, user experiences grow more realistic with innovations like high-resolution displays and immersive audio. VR-based virtual exhibitions, widely used in museums and art fields, offer immersive experiences like *Mona Lisa: Beyond the Glass* at the Louvre Museum[26] and VR tours by the National Museum of Natural History[29] and the Palace of Versailles[30]. Advancements in computer graphics make it easy to model artifacts and architectural structures, allowing for experiences like exploring Hungarian ballet history in a virtual theater[9]. Virtual exhibitions reconstruct lost or damaged artifacts and sites, surpassing physical exhibitions' limitations [33]. Modern VR technology enables accurate tracking and simulation of users' movements, enhancing interaction[17]. VR exhibitions, like the Virtual Museum of the Antikythera Mechanism[3], offer unique interactions impossible in physical exhibitions, attracting younger audiences to cultural history. The immersive nature of VR exhibitions allows users to fully engage without external distractions, as seen in educational VR exhibitions like the Vauquois Tunnel[10]. Virtual exhibitions can integrate with online resources, offering personalized experiences tailored to user preferences, as explored by scholars combining VR with recommendation systems to track user interests[19]. However, to the best of our knowledge, there is a scarcity of virtual reality exhibitions relating to Hanfu.

### 2.2 Virtual Reality in Education

Initially, VR was linked mainly to gaming and entertainment, but its use in education is gaining traction for its ability to create immersive learning environments. Using VR environments to develop serious games, which are designed for education or training, as suggested by Tsita[36] , can enhance learning and enjoyment. Ni[28] points out VR's potential in revolutionizing early childhood education through innovative instructional methods. Parmar[32] evaluates VR's impact on middle school students learning computer programming, suggesting immersive environments and self-avatars enhance education. In higher education, researchers[2, 6, 11, 13, 20] highlight VR's potential to enhance learning, but a study by Slavova[35] emphasizes the need for improved social interaction and productivity tools. Kim[21] introduces a VR MOOC Learning Management System for chemistry experiments, aiming to improve instructional design for interactive VR learning. Kvasznicza [22] develops a VR project to enhance learning experiences for electrical engineering students, aiming to boost academic performance and job prospects. The emergence of VR in education signifies a promising shift in knowledge dissemination, emphasizing its importance as a potent

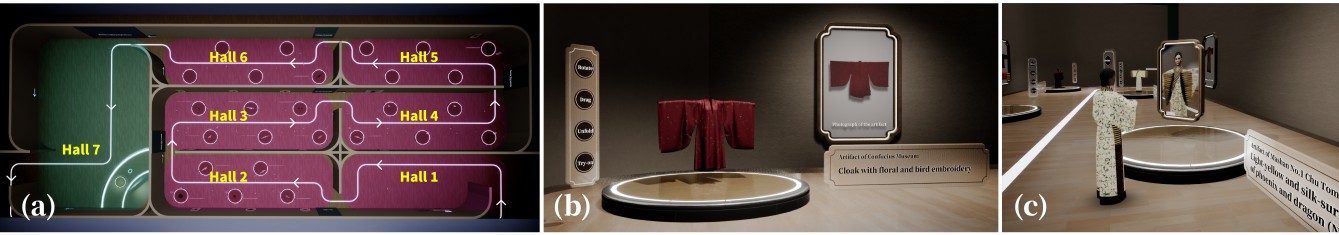

**Figure 3: Virtual exhibition halls in VisHanfu system (a) and the 3D models with artifact photos placed in hall 2-6 (b). Our system provides users with four methods of interacting with the Hanfu artifact model, including: Rotate, Drag, Unfold, and Try-on. (c) shows the scene when the user tries on the model in third-person perspective (the first-person perspective is applied when user experiences)**

.

instructional tool. Consequently, VR could be utilized to promote and popularize Chinese fashion culture.

## 3 SYSTEM DESIGN

### 3.1 Overview

To increase the user's interest in learning and spreading Hanfu culture, we developed the VisHanfu virtual reality system concentrating on the cross-shaped flat structure. The system consists of two parts: digital Hanfu artifact exhibition and Hanfu-making experience (refer to the magenta and cyan parts in Fig. 3(a)). According to reliable historical documents, we described the characteristics and origins of the cross-shaped flat structure in the exhibition part and digitally reconstructed twenty-five sets of Hanfu artifacts from various dynasties for users to observe (one example is shown in Fig. 3(b)). In the Hanfu-making experience section, users are able to create a virtual Hanfu by completing some easy interactive tasks that we have designed. By using two kinds of presentations, i.e., high realistic 3D models and animated dance motion sequence (Section 3.5), users can observe the non-rigid movement of the Hanfu. The four components, i.e., "Virtual Scenario Construction", "Digital Restoration of Hanfu artifacts", "Virtual Hanfu Making", and "Display of the virtual Hanfu" are crucial to the development of our VisHanfu system, which are described in the following sections.

### 3.2 Virtual Scenario Construction

We designed a virtual reality scenario that simulates a real museum, which consists of an exhibition and an experience part (Fig. 3(a)). The exhibition part includes six exhibition halls (exhibition halls 1-6), and the experience part occupies one of the exhibition halls (the exhibition hall 7).

In hall 1, we first introduce the origin and characteristics of the cross-shaped flat structure. Next, we set up five exhibition halls corresponding to the five dynasties of China (Qin, Han, Tang, Song and Ming) in chronological order. This arrangement makes it possible for users to comprehend how Hanfu evolved over many historical periods. In each exhibition hall, not only is a representative 3D model of Hanfu artifacts with cross-shaped flat structure in that era placed, but on the wall beside the model, photographs of the artifacts, diagrams of their cross-shaped flat structure, diagrams of

their patterns, as well as textual introductions are displayed to help the user learn about the artifacts more thoroughly.

### 3.3 Digital Restoration of Hanfu artifacts

Textile artifacts are prone to fading and decay over time. It is difficult to help the public understand the real color, texture, and shape of the artifacts based on the photos of the excavated artifacts. Therefore, we consulted experts in Hanfu research and chose twenty-five sets of Hanfu artifacts representative of different dynasties to be digitally restored, in order to help the audience better understand the Hanfu culture. We first conducted literature research to understand the structure diagrams, size information, materials, pattern diagrams, etc. of the Hanfu artifacts that we aim to restore from the archaeological reports.

*3.3.1 Mesh reconstructed.* The process of creating a mesh of Hanfu artifact involves several steps (See Section 3.6 for our implementation details). First, a 2D pattern of the garment is generated based on archaeological measurements (Fig. 4(a)). Next, the 2D patterns are used to generate 3D pattern meshes in a virtual space (Fig. 4(b)). The 3D patterns are then arranged around an avatar, and each one is connected according to the sewing relationships of the garment (Fig. 4(d)). Once the patterns are in place, cloth simulation is activated, and the 3D patterns gradually drape over the avatar (Fig. 4(e)). Collars and sleeves are added, and the simulation continues until the garment achieves the desired shape (Fig. 4(f)).

*3.3.2 Texture reconstructed.* The steps to reconstruct a Hanfu texture are shown in Fig. 5. Referencing the chrysanthemum pattern (Fig. 5(a)) unearthed from the Song dynasty tomb of Huang Sheng [40], a basic pattern tile was drawn on a square canvas in Photoshop (Fig. 5(b)). The pattern was arranged to be tileable in four directions seamlessly. Then, textures for mask according to distinct colors were exported. The final tiling and textured pattern were created within Substance Painter (Fig. 5(c)). With the garment imported, color selection was then used to separate materials according to the ID map, so that multiple materials could be integrated into one map to save space (Fig. 5(d)). After texturing, ambient occlusion, roughness, and metallic maps were packed together in R, G, B channels of one texture map before exporting, which is an efficient method as it packs 3 monochromatic maps while keeping the quality of

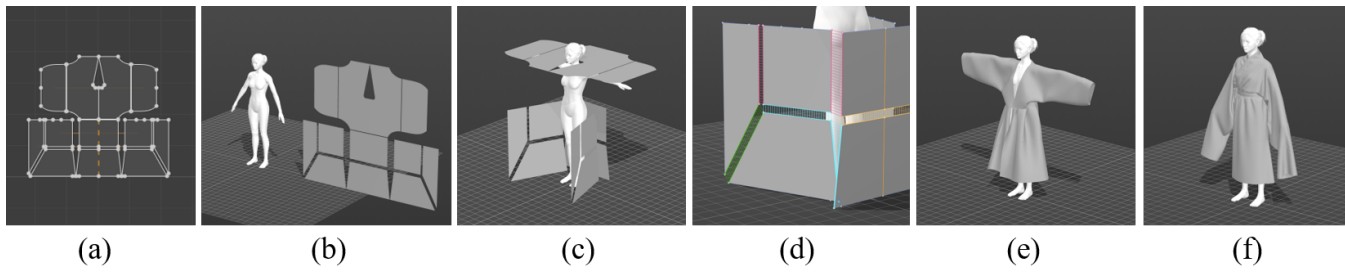

(a)      (b)      (c)      (d)      (e)      (f)

Figure 4: The process of reconstructing a Hanfu mesh. See text for the description of each step.

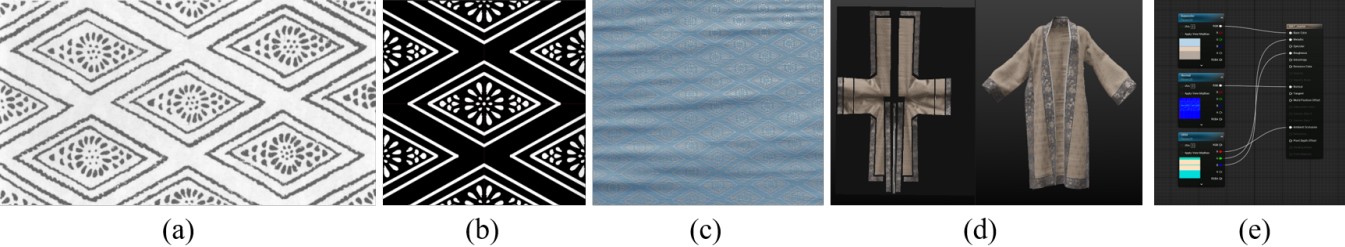

(a)      (b)      (c)      (d)      (e)

Figure 5: The method of reconstructing a Hanfu texture. See text for the description of each step.

the textures. In the UE material editor, each map channel was connected correspondingly and texture maps were parameterized to form a master material for instancing [16] (Fig. 5(e)). By creating material instances and replacing texture maps, a new texture set under the same workflow is able to utilize the same material functions without incurring an expensive recompilation of the material [16]. The 3D models are placed on platforms in each hall (see the white circle in Fig. 3a), and users can use the controller to rotate the model and adjust their viewing distance to enable them to observe the artifacts conveniently. Combined with the illustrations on the side of the model, users can better understand the information on the artifacts and the changes in Hanfu styles in different dynasties.

### 3.4 Virtual Hanfu Making

We have divided the traditional Hanfu-making process (according to the stack-cutting method in literature [42]) roughly into four simple steps. To complete the production of a Hanfu, the user only needs to follow the prompts to complete a series of simple interactive tasks. This part aims to help users to experience and understand the process of making traditional Hanfu. The four steps are described as follows (See Fig. 6) :

**Step1**: Fabric slitting. To divide a whole piece of fabric into two pieces, the user slides the "scissors" icon along the dotted line using the VR controller.

**Step2**: Fabric folding. The user uses the VR controller to manipulate the "hand" icon along the dotted line to fold and stack the pieces cut in Step 1 along the center line.

**Step3**: Cutting. The user cuts the fabric into pieces along the dotted line with the controller by manipulating the "scissors" icon.

**Step4**: Unfolding and Sewing. The user uses the controller to manipulate the "hand" icon to unfold the pieces, and our system

will automatically perform the sewing step to get the complete Hanfu.

### 3.5 Display of the virtual Hanfu

In this part, the virtual Hanfu produced in Section 3.4 is displayed. With the help of cloth simulation technology, the high realistic non-rigid movement of Hanfu is possible. The two display solutions as follows:

(1) 3D model. The 3D model of virtual Hanfu is placed on the platform, similar to the Hanfu artifact display method stated in Section 3.3. A variety of textures are pre-produced for users to choose from (Fig. 7a). By switching procedures, users can view Hanfu with various textures. Using the cloth simulation technology [25] [39] [12], users are able to observe the high realistic non-rigid motion effect by dragging and dropping the model vertices (Fig. 7b). This greatly improves the users' experience and increases the users' freedom of observation of the clothes in comparison to the conventional rigid transformations like translation, rotation, zoom-in, and zoom-out.

(2) Performance of Dance. We invited a experienced dancer to perform a Chinese classical dance, used motion capture technology to record her movement data (Fig. 8), and migrated that movement data to a high-precision MetaHuman model with a typical Chinese face to generate an animation of avatar dancing in Hanfu. This process also uses cloth simulation techniques. Combining Hanfu with Chinese classical dance allows users to better appreciate the aesthetics of Hanfu.

### 3.6 Implementation

Our VisHanfu system is developed using Unreal Engine 5 [15] (UE5, Version 5.1.1) and optimized for the VIVE XR Elite headset [38]. The

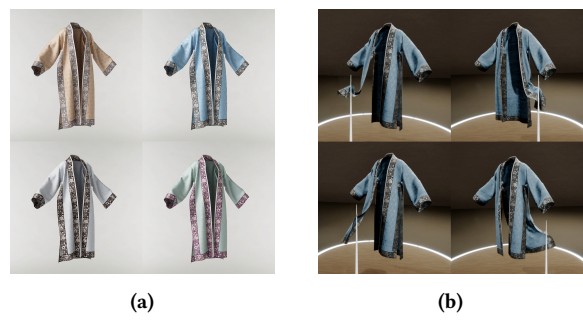

**Figure 6: The process of traditional Hanfu production consists of four steps, including "Fabric slitting", "Fabric folding", "Cutting", "Unfolding and Sewing". Users can interactively experience this process in the VisHanfu system.**

**(a)**     **(b)**

**Figure 7: The display of the Hanfu model. (a) A variety of textures are pre-produced for users to choose from. (b) Users are able to observe the non-rigid motion effect by dragging and dropping the model vertices. See demo video for more details.**

digital restoration garment models were created in Style3D [24] and imported into UE5. To enhance fabric material performance, Adobe Substance 3D Painter [1] was used to generate highly realistic texture bitmaps, which were then imported into UE5 and transformed into material assets. Vicon [37] optical motion capture equipment and Autodesk Motion Builder [4] were used to produce dance animation compatible with the Metahuman avatar [14].

## 4 USER STUDY

We conducted two user studies in a local museum and in the laboratory to find out how users learned about Hanfu knowledge and their perspectives on the system.

### 4.1 User study 1 (in the Museum)

We deployed our system in a local museum for a week and recruited visitors to use our system and give feedback.

*4.1.1 Participants.* There were 150 participants, including 69 males and 81 females, who used our system, ranging in age from 8 to 68 years old ($M = 19.15$, $SD = 11.86$). Participants were recruited based on the following criteria: (1) Participants must be able to wear the VR glasses properly, so we excluded children whose head circumference was too small for it. (2) Participants must have normal or corrected-to-normal vision and be able to see the VR scenario clearly.

*4.1.2 Procedure.* Before the experiment began, the experimental procedures were explained to participants, and informed consent was obtained from each individual. After ensuring that they could operate the system correctly, users used the VisHanfu system to learn about the cross-shaped flat structure, observe Hanfu artifact models, and experience Hanfu making. After the VR experience, users completed a 11-question questionnaire and were interviewed to report on their feelings about using the VR system.

*4.1.3 Questionnaire.* The questionnaire contains 11 questions, of which 1-10 are from the System Usability Scale (SUS) [7], which is used to evaluate the usability of the system. Users rated each of the 10 questions according to their level of agreement on a scale of 1 to 5, with 1 being strongly disagree and 5 being strongly agree. Question 11 is designed to investigate whether the user's interest in learning Hanfu culture has been enhanced after using the system.

*4.1.4 Results of questionnaire.* We report the questionnaire results of the user study at the museum in this section, and the results

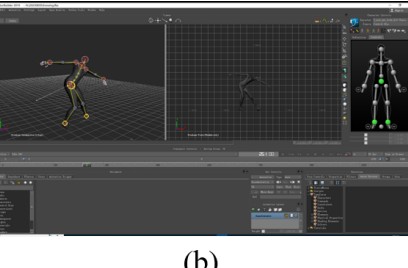 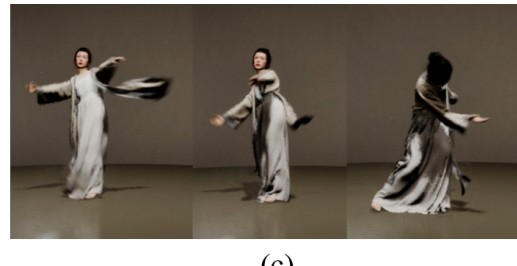

(a)     (b)     (c)

**Figure 8: The generation process of an animation with avatar dancing in Hanfu. (a) Motion capture. (b) Data migration. (C) Screenshots of the animation. See demo video for animation details.**

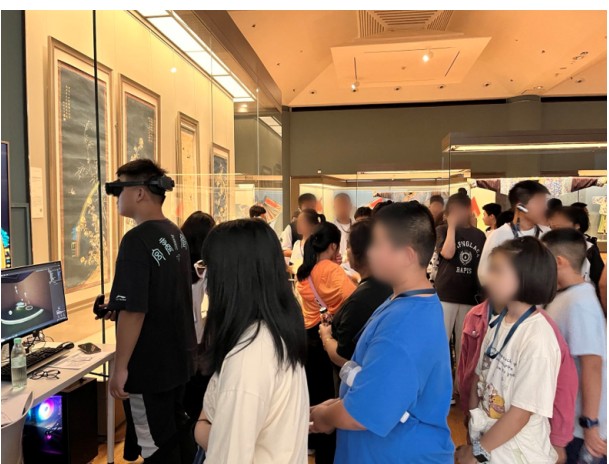

**Figure 9: User study 1 in a local museum.**

of the interview will be reported in section 4.3. The questionnaire results indicate that our system has good usability and helps users increase their interest in learning about Hanfu culture.

According to the scoring rules of the SUS scale, the raw data of the SUS scale (Question 1 to 10) was processed as follows: (1) For the questions with an odd number, subtract 1 from the score. (2) For the questions with an even number, subtract their value from 5. (3) Add up new values to obtain the total score. (4) Multiply the total score by 2.5.

It returns a score of 79.80/100 with an SD of 11.48, which indicates VisHanfu system has good usability [5].

We also explored users' subjective perceptions of whether their interest in learning about Hanfu culture increased after using the system through Question 11. A score of 5 means "strongly agree" that the VisHanfu system enhances user's interest in Hanfu culture, and a score of 1 means that the system is not useful at all in enhancing user's interest. The mean score for this question was 4.60 with a standard deviation of 0.73. We further counted the corresponding numbers of people with different scores and the results showed that more than 98% of the participants thought that our system could help enhance their interest in learning Hanfu (scoring greater than or equal to 3).

## 4.2 User study 2 (in the Laboratory)

In addition to the usability of the system and the effectiveness of increasing user interest studied in user study 1, we would like to further investigate the effectiveness and engagement of the VisHanfu system, then we conducted user study 2 in the lab.

*4.2.1 Participants.* We recruited 60 participants, 30 in each of the experimental and control groups. The experimental group included 14 males and 16 females, ranging in age from 19 to 33 ($M = 23.80$, $SD = 3.80$). The control group included 15 males and 15 females, ranging in age from 19 to 35 ($M = 25.93$, $SD = 4.22$). Participants were recruited based on the following criteria:: (1) Participants must be able to wear the VR glasses properly, so we excluded participants whose head circumference was too small for it. (2) Participants must have normal and corrected-to-normal vision and be able to see the VR scenario clearly.

*4.2.2 Procedure.* The experimental procedures for the experimental and control groups are as follows:

**Experimental group.** Participants in the experimental group received instructions on how to use the system, and once it had been confirmed that they can use it properly, they were asked to utilize VisHanfu for around 20 minutes to learn about cross-shaped flat structures, view twenty-five Hanfu artifacts, and experience the cutting process. After that, the user will answer a 28-item questionnaire and receive an interview.

**Control Group.** Participants spent 20 minutes studying a paper-based material that contained all the Hanfu-related knowledge mentioned in the VisHanfu system, including the origins and characteristics of the cross-shaped flat structure, illustrations and textual descriptions of the twenty-five Hanfu artifacts, and the Hanfu production method. A 5-item questionnaire was completed by the participants at the end of the study.

*4.2.3 Questionnaire.* There were differences in the questionnaires of the two groups, which we describe below.

**Experimental Group.** The experimental group questionnaire consisted of 28 questions divided into two parts. The first part contained 23 questions that allowed the users to subjectively score the statements on a scale between 1 and 5 (1=strongly disagree, 5=strongly agree). Questions 1-11 were the same as the questionnaire in user study 1 (Questions 1-10 were from the SUS scale, and question 11 was used to investigate whether the user's interest in

**Table 1: The results of the engagement evaluated by UES-sf scale.**

|  | Overall | Focused attention | Perceived usability | Aesthetic appeal | Reward |
|---|---|---|---|---|---|
| Score | 4.26 | 4.89 | 3.53 | 4.42 | 4.58 |

learning Hanfu has increased after using the system). Questions 12-23 were from UES-SF scale for assessing the users' engagement. The second part contained five questions (Question 24-28) about Hanfu culture, which were used to assess the effect of using the VisHanfu system to learn Hanfu knowledge.

**Control Group.** The questionnaire for the control group contained 5 questions, the same as the second part of the questionnaire for the experimental group.

*4.2.4 Questionnaire results.* The questionnaire results of user study 2 are reported below. According to the results, our system has good usability, helps to enhance users' interest in learning about Hanfu culture, and performs well on scales assessing user engagement. In addition, the experiment demonstrated that users studying Hanfu-related knowledge using our system (experimental group) scored higher on the Hanfu-related knowledge questionnaire compared to participants using traditional reading materials (control group).

**System Usability (Question 1-10):** We calculate the SUS score as described in Section 4.1.4, and it returns a score of 75.00/100 with an SD of 13.82, which indicates VisHanfu system has a good usability [5].

**User Interest (Question 11):** We also investigated users' subjective opinions of whether their interest in studying Hanfu culture has risen as a result of utilizing the system. A score of 5 means "strongly agree" that the VisHanfu system increases their interest in Hanfu culture, while a score of 1 means that they do not think the system contributes anything to increase their interest. The mean score for this question was 4.40 with a standard deviation of 0.68. The number of individuals with various scores was also counted. Four out of twenty people scored a 3, twelve scored a 4, and fifteen scored a 5. The findings revealed that, for all participants (who scored greater than or equal to 3), our system might increase their interest in learning Hanfu culture.

**User Engagement (Question 12-23):** We use the UES-sf scale for measuring users' engagement in using the VisHanfu system, and it is further divided into four subscales to investigate the focused attention (FA, Question 12-14), perceived usability (PU, Question 15-17), aesthetic appeal (AE, Question 18-20), and reward (RW, Question 21-23). According to the calculation rules in [31], the results of UES-sf scale are shown in Table 1. From the results, it is demonstrated that the VisHanfu system has achieved higher ratings, both overall and on the subscales.

**Learning Effectiveness (Question 24-28):** These questions were used to evaluate the effectiveness of learning Hanfu knowledge in different ways. Five questions were included. The user's knowledge of the Hanfu artifacts displayed in the exhibition part was assessed using Questions 24 and 25, while understanding of the characteristics and origins of the cross-shaped flat structure was assessed using Questions 26 and 28, respectively. We designed

**Table 2: The accuracy (in percentage) of Question 24 to 28 in both experimental and control groups.**

| Question | 24 | 25 | 26 | 27 | 28 |
|---|---|---|---|---|---|
| Experimental group | 86.67 | 93.33 | 80.00 | 100.00 | 100.00 |
| Control group | 83.33 | 83.33 | 53.33 | 66.67 | 80.00 |

Question 27 to evaluate the user's comprehension of the Hanfu production process.

We calculated the accuracy for each question in the experimental and control groups (Table 2). We performed the Chi-squared test to explore whether the accuracy between the two groups on each question was significantly different. On Question 24, the accuracy of the experimental group is 86.67%, and the control group is 83.33%. On Question 25, the experimental group's accuracy is 93.33%, while the control group has only 83.33%. We do not observe a significant difference between the two groups on Question 24 ($\chi^2 = 0.13$, $p = 0.72$) and Question 25 ($\chi^2 = 1.46$, $p = 0.23$). Question 26 was more difficult, so the accuracy of both groups was lower, 80% for the experimental group and 53.33% for the control group. There was a significant difference between the two groups on this question ($\chi^2 = 4.80$, $p = 0.03 < 0.05$). Questions 27 and 28 reached 100% correctness for the experimental group, whereas for the control group, the accuracies were only 66.67% and 80% respectively. And significant differences were observed on these two questions (Q27: $\chi^2 = 12.00$, $p = 0.001 < 0.05$ ; Q28: $\chi^2 = 6.67$, $p = 0.01 < 0.05$). It shows that on all five questions, the experimental group obtained a higher accuracy than the control group.

The total score of Questions 24 to 28 was analyzed using the independent-samples T-test, and there are significant differences between the experimental and control groups ($t = 4.16$, $p = 0.000 < 0.05$). The results indicate that using the VisHanfu system allowed users in the experimental group to more thoroughly understand knowledge about Hanfu, including the origins and characteristics of the cross-shaped flat structure, information about the twenty-five Hanfu artifacts, and the method of Hanfu production.

## 4.3 Results of interview

In addition to the questionnaire, we further inquired about the user's feelings towards the system through an interview (in both user studies), asking them to answer the following three questions as well as provide any additional perspectives.

*MQ1: Has there been an increase in interest in learning about Hanfu culture? Will you take the initiative to learn about it afterward? How do you learn about it?*

We realized that a subjective score based on Question 11 of the questionnaire is not enough to demonstrate that their interest in Hanfu culture has indeed increased and that this conclusion can only be reached after a long period of follow-up. Accordingly, we would like to further analyze this issue by using the question *MQ1* in the interviews.

We found that those who reported an increased interest in Hanfu said that they had taken the initiative to learn more about Hanfu through, but not limited to, actively reading books, searching the web, and traveling to museums. Some children demonstrated a strong interest in learning by giving a clear time schedule: "I will

ask my mother to take me to a museum with more Hanfu artifacts on my next holiday". Some participants actively asked us about Hanfu during the system experience, for example, "Where was this artifact unearthed?", "Who wore it before?", "On what occasion was this suit worn?" In addition, after the experience, some participants took the initiative to search for issues of interest through their mobile phones and tried to discuss them with our staff, for example, "Was this dress in this colour in ancient times?", "What is the name of this cutting method?" Based on this feedback, we think that the VisHanfu system has increased the interest of some users in learning about Hanfu culture and therefore has the potential to promote it.

*MQ2: What is the difference between using the VisHanfu system to learn Hanfu culture and the traditional way of learning? Which one do you prefer?*

This question was used to help us understand the advantages and disadvantages of this interactive virtual reality-based system compared to traditional learning methods such as reading, visiting, and watching videos. We summarized the users' responses and found that they generally found this novel interactive system more appealing for them to learn about Hanfu culture than the traditional common learning methods. Our system is more immersive and intuitive than reading written materials and watching videos. One boy said, "This system makes me more willing to explore on my own initiative." Almost all the children said they preferred this type of interactive learning. Moreover, some parents said that using this immersive VR system makes little difference to their children compared to taking them to a museum to see artifacts, but minimizes their travel costs, making it very friendly to them.

## 5 DISCUSSION

Through two user studies, we found that age affects users' use and evaluation of the VisHanfu system.

Children were more willing to experience the VisHanfu system than adults. User study 1 was conducted in a local museum, and since it was during the summer holidays, there would be a lot of children as well as their parents. However, we found that the number of adults who were willing to experience it was far less than that of children. Only a few adults came forward to inquire whether they could participate, but the vast majority of them wished to let their children get the opportunity rather than themselves, and they would explain that they had not used a VR device before or did not understand the operation of the VR device, thus refusing our invitation. We could only entice them to give feedback on the experience by explaining that it was very simple to operate. Although the minimum age of the users reported in the user study was 8 years old, there were actually some children younger than 8 years old who also expressed a strong willingness to use the device, but due to their head circumference being too small to wear the glasses properly, they could not complete the experience on their own, so these participants were excluded from the report on the results of the user study. Some of the children were willing to queue up several times after the experience to try it again, even though it took them a lot of time.

Children often tended to give more positive evaluations, which was also reflected in the scoring of the usability scale. Compared to

a noisy museum that can be disturbed by others (which is why we did not test user engagement and learning effect in user study 1, where others also interacted with the user during their experience, which would have affected both results), a quiet lab allowed the user to be more attentive to learning the system's instructions and operating it, and we would have expected that the SUS scales in user study 2 scores would be higher. However, the results show that user study 1 scored 79.8 on the SUS scale, while user study 2 scored 75.00. This may be due to the fact that the users in user study 1 were more likely to be children, who tended to give more positive ratings.

There were differences between the focus of adults and children. We found that children were more concerned about the richness of the content presented by the system, and they tended to gain more knowledge. For example, many of them expressed their wish to have more content to learn (e.g., more models of artifacts, different ways of making Hanfu, etc.), but adult users were more concerned about whether the operation is convenient, the instructions are clear, and the experience is smooth. Synthesizing their views can help us make enhancements to the system.

## 6 LIMITATION AND FUTURE WORK

The system was not specially designed for users of different ages, so through the user study, we found that the Hanfu-making experience part of the system was relatively simple for middle-aged people, although the difficulty was appropriate for children and the elderly. In order to provide users the freedom to pick the level of difficulty of the experience, we will further update this part by adding different modes (for example, the procedures for making the Hanfu in the HARD mode will be split in more detailed steps).

## 7 CONCLUSION

Hanfu is the traditional costume of Han nationality in China, involving textile, embroidery and other handicrafts, which is very important for the inheritance of Chinese traditional culture. Aiming at the problems of single experience and lack of convenience in existing promotion methods, we designed the VisHanfu virtual reality system, which allows users to observe the exquisite Hanfu artifacts and experience the Hanfu making process, to enhance the users' interest in learning Hanfu culture, and to help the spread of it. We propose a solution for the digital reconstruction of Hanfu artifacts. Compared to the existing methods, our approach achieves an improvement in the efficiency of texture reconstruction by parameterizing the texture maps while ensuring accuracy. In terms of Hanfu model display, our system incorporates cloth simulation to help users observe the real-time non-rigid motion of their created Hanfu through drag-and-drop operations and captures the dancer's movement data so that users will be able to watch an avatar perform a Chinese dance in their created Hanfu. Two user studies were conducted. One user study with 150 people in a museum showed that our system has good usability and is helpful in increasing users' interest in learning Hanfu culture. The other user study with 60 people in a laboratory not only obtained similar conclusions as the previous user study but also further showed that our system helped users gain a good sense of engagement.

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
