# OpenReview forum: "VisHanfu: An Interactive System for the Promotion of Hanfu Knowledge via Cross-Shaped Flat Structure"
_acmmm.org/ACMMM/2024/Conference — MM2024 Poster_

### Official Review · Reviewer_AQEE · 2024-05-04

**Rating:** 5
**Confidence:** 2

**Summary:**

In this paper, the authors developed an Interactive System for Hanfu exhibition and Hanfu experience based on VR device.
The developed system consists of two parts: digital Hanfu artifact exhibition and Hanfu-making experience, which may help the participants know the history of Hanfu culture, and make them more interested in Hanfu culture. The experimental results show that the interactive system does prompt the Hanfu culture.

**Strengths:**

- This work is interesting and meaningful, which combines the traditional Hanfu-making process and the new VR technology. In my opinion, this work indeed helps the Hanfu culture promotion.
- The details of the interactive system are illustrated well.

**Limitations:**

- The experimental results are not illustrated in a good way. More tables and figures may be better.
- Did the author analyze the influence of the participants' different backgrounds on the results of the experiments?
- The experimental results show that some participants prefer to join again. Did the author analyze whether this is due to children preferring VR experiences or being attracted by the Hanfu?

**Suitability:**

2

---

### Official Review · Reviewer_iB8i · 2024-05-19

**Rating:** 5
**Confidence:** 3

**Summary:**

The paper presents VisHanfu, an VR system that can be used in museum for education. While it mainly focused on one feature, the crossed shaped flat structure, it showcases the potential of VR for public education. Two user studies were conducted, and the results are consistent.

**Strengths:**

While the idea of enabling public education via VR is nothing new, the VR system built in this paper demonstrated the potential of such an approach.

**Limitations:**

The approach is very focused on this particular scenario. Whether and how this can be extended to more generic scenarios could use some discussions.

The analysis of user studies could be more detailed.

**Suitability:**

3

---

### Official Review · Reviewer_73BJ · 2024-05-24

**Rating:** 3
**Confidence:** 2

**Summary:**

The VisHanfu system addresses these issues by focusing on the cross-shaped flat structure of Hanfu. It digitally restores twenty-five representative Hanfu artifacts, allowing users to interact with and experience the Hanfu-making process. The system combines realistic cloth simulation techniques with interactive tasks, enabling users to observe the fluid movements of Hanfu garments and participate in the making process.

**Strengths:**

The VisHanfu system's emphasis on interaction and immersion distinguishes it from traditional methods of cultural dissemination. The ability to digitally restore Hanfu artifacts and allow users to interact with them is a novel application of VR technology. Some important findings are as follows:

1. The theoretical discussion on combining cultural heritage preservation with advanced VR technology offers a framework and artwork.

2. The paper provides a comprehensive description of the system design, including the digital restoration of Hanfu artifacts, virtual Hanfu making, and realistic cloth simulation. This detailed technical explanation ensures the system is both functional and aligned with traditional craftsmanship.

3. The phases of the two user studies are very well developed, and the number of participants can very well support the effectiveness of this system.

**Limitations:**

The paper glosses over the potential inaccuracies in the digital reconstruction of Hanfu artifacts. Without stringent validation by cultural experts, the system risks perpetuating incorrect representations of Hanfu, which can mislead users and disrespect the cultural heritage it aims to preserve. It is possible to incorporate expert evaluation into the user study, especially from the perspective of experts (such as physical designers of Hanfu/inheriting intangible cultural heritage/museum staff) to supplement the user study.

**Suitability:**

2

---

### Official Review · Reviewer_gQS5 · 2024-05-25

**Rating:** 1
**Confidence:** 4

**Summary:**

The article contributes to the community by presenting a VR system whose main objective is to address the issue of Hanfu with the population and conserve their culture.
In parallel we have the issues: digital restoration, immersion and learning with VR and finally cultural promotion.

**Strengths:**

The article is well written, has visual images and supplementary files with great instructions.
It has good structural formatting and shows an evaluation study.

**Limitations:**

The article presents good theoretical content and is attractively written. However, there is no mention of the ethics committee process number, nor data to show that this is a unique research and not a continuation of the already published article https://dl.acm.org/doi/pdf/10.1145/3613905.3651114

**Suitability:**

3

---

### Meta-Review · Area_Chair_sRfJ · 2024-07-02

**Recommendation:** Accept (Poster)
**Confidence:** 4

**Metareview:**

The paper introduces the VisHanfu system, a VR application aimed at preserving and promoting Hanfu culture through digital restoration, immersive interaction, and educational experiences. The system allows users to engage with Hanfu artifacts and participate in the Hanfu-making process, focusing on realistic cloth simulation and interactive tasks. The work includes evaluations through user studies to demonstrate the system's effectiveness.

The paper has received mixed feedback from the reviewers but is leaning towards acceptance. They have identified several strengths:
The system's focus on combining cultural heritage preservation with advanced VR technology offers a novel approach to cultural dissemination. The paper comprehensively describes the system design, including digital restoration, virtual Hanfu making, and cloth simulation, ensuring clarity and functionality. The system has been tested practically, adding credibility to its utility and performance claims. The phases of the two user studies are well-developed, with a sufficient number of participants to support the system's effectiveness.

While the paper is well written and the approach is innovative, it is important to focus on areas for improvement by addressing current limitations highlighted by the reviewers.

1. The revision must address ethical considerations as detailed in the author's rebuttal.
2. The digital reconstruction of Hanfu artifacts may have inaccuracies without validation by cultural experts, risking misrepresentation.
3. The approach is highly specific to Hanfu, and discussions on extending the system to more generic scenarios are limited.
4. The analysis of user studies could be more detailed, particularly regarding the influence of participants' backgrounds on the results.

Based on the reviews, the paper is recommended to be accepted as a poster presentation. The design process (theoretical discussion) and the VisHanfu system present a valuable contribution to cultural heritage preservation through the innovative use of VR technology and justify acceptance as a poster.